# Prevalence of peripheral arterial disease and arterial calcification based on three ankle-brachial index calculation methods (highest, average, and lowest systolic ankle pressure): A cross-sectional study in Type 2 diabetes mellitus patients in Peru

Luis Fernando Espinoza-Enciso[1], Iván Gonzalo Hernández-Gozar[1], Kevin Clared Zuñiga-Baldarrago[1], Robert Lozano-Purizaca[2,] Manolo Briceño-Alvarado[3,4]*, Marlon Yovera-Aldana [5]

**1** School of Medicine, Universidad Científica del Sur, Lima, Perú, **2** School of Medicine, Universidad Nacional de Piura, Piura, Perú, **3** Grupo de Investigación MBA Cardiovasculares, Lima, Perú, **4** Villamedic Groups, Lima, Perú, **5** Grupo de Investigación de Neurociencias, Metabolismo y Efectividad clínica y Sanitaria, Universidad Científica del Sur, Lima, Perú

* myovera@cientifica.edu.pe

## Abstract

### Background

Peripheral arterial disease (PAD) and arterial calcification (AC) are frequent yet underdiagnosed vascular complications in individuals with type 2 diabetes mellitus (T2DM). The ankle-brachial index (ABI) is a widely used, non-invasive too for detecting these conditions. However, differences in ABI calculation methos can impact diagnostic accuracy and prevalence estimates.

### Objective

To determine the prevalence of PAD and AC based on three ABI calculation methods in patients with T2DM attending a public hospital in Peru.

### Methodology

We conducted a cross-sectional study using data from the At-Risk Foot Program of the Endocrinology Department at Hospital María Auxiliadora (2015–2020). ABI was calculated for each lower limb using the highest, average or lowest systolic ankle pressure (SAP) from either the dorsalis pedis or posterior tibial artery as the numerator, divided by the highest brachial systolic pressure as the denominator. We applied a hierarchical classification: PAD was identified first (ABI < 0.9 in either limb), and among those without PAD, AC was identified (ABI > 1.3 in either limb); the rest were classified as normal. Prevalences estimates were calculated with 95% confidence

**Data availability statement:** All relevant data are within the paper and its Supporting Information files.

**Funding:** The author(s) received no specific funding for this work.

**Competing interests:** The authors have declared that no competing interests exist.

intervals, and associations with clinical characteristics were explored using Poisson regression with robust variance.

## Results

We included 643 subjects with a mean age of 61.4 years, 69.8% female. The prevalence of PAD was 7.8% (95% CI: 5.8–10.1), 15.4% (95% CI:12.7–18.4), and 28.2% (95% CI 24.7–31.7) using the highest, average, or lowest SAP as the numerator in the ABI, respectively. Conversely, the prevalence of AC was 18.2% (95% CI: 15.3–21.4), 11.0% (95% CI: 8.7–13.7), and 16.2% (95% CI:13.4–19.3). In all three methods, PAD was associated with older age (p < 0.05) and AC was associated with longer duration of diabetes (p < 0.01).

## Conclusions

Among patients with T2DM, PAD prevalence varied substantially (7.8% − 28.2%) depending on the ABI calculation method, while AC was present in up to 18.2%. The lowest ankle pressure method increased sensitivity and may be preferred in high-risk populations where avoiding missed diagnoses is critical. The highest ankle pressure method, which maximizes specificity, may be more suitable for general screening and comparability with existing literature, whereas the average pressure approach could be useful in research or prognostic modeling. The hierarchical classification strategy allowed PAD and AC to coexist in the same individual, although this was rare. Given the variability in prevalence across methods, local validation studies are needed to determine which approach optimally balances sensitivity, specificity, and clinical applicability in Peruvian diabetic populations.

## Introduction

Atherosclerotic diseases continue to progress globally due to non-communicable conditions such as diabetes mellitus, hypertension, and dyslipidemia, exacerbated by increased life expectancy and unhealthy lifestyles [1]. These diseases are the leading cause of death in high, middle, and low-income countries [2]. Among them, peripheral arterial disease (PAD) is particularly concerning due to its strong association with lower limb amputation, especially in patients with diabetes mellitus (DM) [3]. In these patients, PAD often presents with bilateral, infrainguinal lesions and medial arterial calcifications and is frequently asymptomatic due to peripheral neuropathy [4], which complicates diagnosis, management and follow-up [5].

PAD affects 11.2% to 14.3% of people with diabetes globally [6]. Critical limb ischemia contributes to nearly 30% of major amputations, with up to 20% of affected individuals dying within six months [7]. In Peru, PAD is often underdiagnosed due to limited systematic screening [8]. While some hospital-based study have reported high prevalence in patients with diabetic foot, few have used standardized ABI measurement [9,10]

Initial diagnostic tools include clinical staging systems like Rutherford and Fontaine [11]. However, many patients with DM, lack typical claudication symptoms due to peripheral neuropathy [12]. Therefore the Ankle-Brachial Index (ABI) has become a widely used, non-invasive diagnostic tool. Traditionally, ABI is calculated by dividing the highest systolic ankle pressure (either from the dorsalis pedis or posterior tibial artery) by the highest brachial systolic pressure. An ABI < 0.9 considered abnormal while an ABI > 1.3 suggests arterial incompressibility due to medial calcification. In patients with diabetes, however, the traditional method may have limited sensitivity due to the presence of arterial stiffness or calcified vessels, [13]although specificity remains high (89.3%). [14].

To improve detection of subclinical or early-stage PA, alternative methods have been proposed using the lowest or the average systolic ankle pressure in the numerator [15]. These approaches aim to balance diagnostic sensitivity and specificity while enhancing the prediction of cardiovascular risk and lower-limb function. [16]Although the highest ankle pressure remains the standard for population studies due to its specificity and comparability, the lowest pressure improves sensitivity for detecting milder or asymptomatic cases. [17] The average pressure has shown stronger associations with functional performance, such as walking speed and leg strength. [18] Each method yields different PAD prevalence estimates and identifies distinct risk profiles.

Additionally, many studies on PAD exclude patients with arterial calcification, despite the clinical importance of this condition, particularly in populations with poor metabolic control [19]. In Peru, studies report that only 2% to 10% of patients attending outpatient clinics achieve adequate control of glucose, blood pressure, and lipids levels [20,21]. Recent recommendations to adopt alternative ABI calculation methods that capture milder or asymptomatic cases further highlight the need to assess the full spectrum of vascular alterations [15]. However, there is limited evidence comparing the performance of these methods in Latin American diabetic populations, particularly in Peru, where vascular complications remain under-recognized.

Therefore, this study aims to determine how the prevalence of peripheral arterial disease and arterial calcification varies according to the method used to calculated the ankle-brachial index (highest, average or lowest ankle systolic pressure) in patients with type 2 diabetes mellitus at a public hospital in Peru.

## Materials and methods

### Study design and clinical setting

Across-sectional study was conducted at Hospital María Auxiliadora, located in Lima, Peru. This study represents a secondary analysis of de-identified clinical data collected during diabetic foot consultations as part of the At-Risk Foot Program, led by the Endocrinology Department. The database comprised patients with type 2 diabetes mellitus (T2DM) evaluated between December 1st, 2015, and January 31st, 2020.

### Population and sample

The database included patients with a confirmed diagnosis of diabetes who who underwent ABI measurement as part of routine clinical evaluation. nclusion required the availability of four ABI measurements per patient—two per leg, corresponding to the dorsalis pedis and posterior tibial arteries. Patienst were excluded if they presented with any of the following conditions: active foot ulcers or wounds, severe lower-limb pain preventing cuff placement, poorly controlled blood pressure at the time of assesment, or marker lower-limb edema that could compromise the accuracy of ABI measurement. Individuals with more severe disease manifestations—such as foot ulcers or edema—are systematically excluded, potentially leading to an underestimation of PAD and AC prevalence in the broader diabetic population.

A census sampling approach was applied, incorporating all eligible cases.

### Variables

Peripheral arterial disease (PAD) and arterial calcification (AC): Definitions of PAD and AC in this study were based on the ABI, calculated as the ratio of ankle to brachial systolic pressures. The highest systolic brachial pressure was used as the

denominator in all cases. For the numerator, three separate methods were applied using systolic ankle pressures (SAP) from the same lower limb. (1) the highest SAP, defined as the greater value between the dorsalis pedis and posterior tibial arteries; (2) the lowest SAP, corresponding to the lesser of the two; and (3) the average SAP, calculated as the arithmetic mean of the dorsalis pedis and posterior tibial measurements. [22]For classification, each participant's two ABI values (one per limb) were first evaluated for the presence of PAD; if either ABI met the PAD criterion (ABI < 0.9), the patient was classified as having PAD. Among the remaining participants, those meeting the AC criterion (ABI > 1.3) were classified as having arterial calcification. The rest were classified as normal. [22,23] This hierarchical structure was applied separately for each of the three ABI calculation criteria (Fig 1).

Other variables included demographic, clinical, and laboratory characteristics. Demographic variables were sex (male/female), age group (<60 years, 60–74.9 years, and ≥75 years), and education level (illiterate, elementary, high school, and college). Past medical history variables comprised duration of diabetes (<10 years, 10–19.9 years, and ≥20 years), type of diabetes treatment (none, oral agenst, insulin combined with oral agents, and insulin only), history of foot ulcers (no/yes), diabetic retinopathy (no/yes), coronary artery disease (absent/present), and hypertension (absent/present). Peripheral neuropathy was defined as a Michigan Neuropathy Screening Instrument score >2, painful neuropathy as a Total Symptom Score >2, and chronic kidney disease as an estimated glomerular filtration rate <60 ml/min/1.73 m². The presence of decreased dorsalis pedis or posterior tibial pulse was recorded as absent or present. Body mass index (BMI) was categorized as <25.0 kg/m², 25.0–29.9 kg/m², or ≥30.0 kg/m². Laboratory variables included fasting glucose, HbA1c, creatinine, hemoglobin, HDL cholesterol, LDL cholesterol, triglycerides, and albuminuria. Poor glycemic control was defined as HbA1c ≥ 7%, following the American Diabetes Association guidelines. [24]. Elevated systolic blood pressure was defined as ≥140 mmHg. Lipid targets were defined as LDL cholesterol <100 mg/dL, HDL cholesterol ≥40 mg/dL in

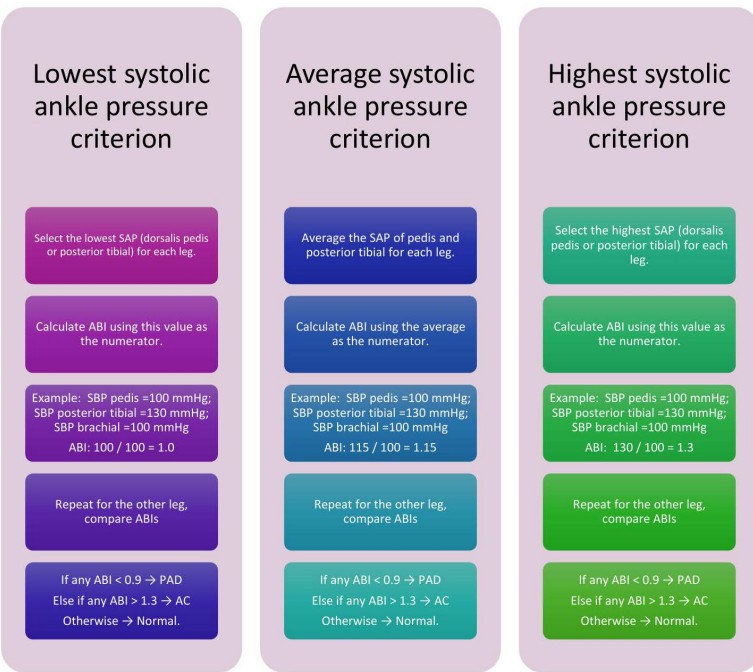

SAP: Systolic ankle pressure. ABI: Ankle brachial index. PAD: Peripheric arterial disease. AC: Arterial calcification

**Fig 1. Diagnostic flowchart for PAD according to ABI criteria.**

men or ≥50 mg/dL in women, and triglycerides <150 mg/dL. Albuminuria was classified as ≥30 mg/24 h. Optimal metabolic control was defined as simultaneous achievement of HbA1c < 7%, LDL cholesterol <100 mg/dL, and systolic blood pressure <140 mmHg. [25].

## At-risk foot program

The At-Risk Foot Program followed the guidelines of the 2015 International Working Group of Diabetic Foot. The clinical assessment included evaluation of symptoms using the Total Symptom Score, review of foot care habits, inspection for deformities, assessment of protective sensation, proprioception, and Achilles tendon reflexes, as well as verification of distal pulses.

The ankle-brachial index (ABI) was measured using a standardized clinical protocol. All assessments were conducted with the patient in the supine position after a rest period of at least 10 minutes. A Yuwell mercury column sphygmomanometer and an 8 MHz Huntleigh handheld Doppler device were used for all measurements. Appropriate cuff size was selected based on limb circumference to ensure accurate readings. Systolic pressures were measured sequentially at the right and left brachial arteries, followed by the dorsalis pedis and posterior tibial arteries of both ankles. The highest systolic brachial pressure was used as the denominator in all ABI calculations. For the numerator, separate readings were taken at each ankle artery; the measurement sequence (right arm, left arm, right ankle, left ankle) was followed consistently to minimize variability.

As part of routine clinical practice, only one measurement per artery was recorded during each evaluation. Although no formal assessment of intra- or interobserver agreement was performed, all evaluations were conducted by two endocrinologists with over 10 years of experience in vascular assessment and diabetic foot care, which supports the reliability of the measurements within the context of standardized clinical procedures.

## Procedures

Permission was requested from the Head of the Endocrinology Service for authorization to use the data. The database was downloaded without names or personally identifiable information. Data were evaluated and analyzed from March 1 to June 30, 2024. The authors filtered records according to the eligibility criteria established. A cleaning of the database was performed to search for extreme data. No imputation was conducted in the case of missing data.

## Analysis plan

The dataset was reviewed in Microsoft Excel to identify extreme values, incomplete data, or inconsistencies. No imputation was performed for missing data. The final analysis included only records with complete ABI measurements in both limbs. Reasons for exclusion due to incomplete ABI data (e.g., edema, hypertensive urgency, acute foot ulcer, or unknown cause) were documented. A brief comparison of selected variables between included and excluded cases was performed.

Data analysis was performed using, STATA® V18 software (Stata Corp, College Station, TX, USA).

Descriptive statistics were computed for all variables. Categorical variables were described using absolute and relative frequencies. Quantitative variables were summarized using means and standard deviations for normality distributed data. or medians and interquartile ranges when normality was no met. The Shapiro-Wilk test was applied to determine distributional assumptions.

The prevalence of PAD, AC, was calculated for each ABI method, along with 95% confidence intervals, using the exact binomial method. To evaluate whether PAD and AC classifications varied significantly depending on the ABI method used, we applied Bowker's test of symmetry for paired ordinal data. This test is a generalization of the McNemar test and is suitable for comparing the symmetry of 3 × 3 classification tables. For each ABI method, the distribution of PAD and AC was described according to demographic, clinical, and laboratory variables.

To identify associated factors for PAD and AC compared to normal ABI, we performed Poisson regression with robust variance to estimate crude prevalence ratios (PR) and 95% confidence intervals. This method was chosen over logistic regression because the outcomes were common, and odds ratios would have overestimated the association. Log-binomial models were initially considered but presented convergence issues. [26,27].

Candidate variables for the multivariable models were selected from the bivariate analysis using a p-value threshold of < 0.20, a commonly accepted exploratory criterion in clinical epidemiology to avoid omitting potentially relevant confounders. [28] The final model reported adjusted prevalence ratios (aPRs). Collinearity was assessed using the uncentered variance inflation factor (VIF) option, and variables with VIF values greater than 10 were excluded. Additionally, variables used in the definition of PAD (e.g., absence of pulse) were not included in the adjustment to prevent collinearity. For models that included variables with incomplete or missing data, a complete-case analysis was performed, restricting the analysis to participants with non-missing values for all covariates. Consequently, the sample size for these models was reduced. No formal comparison was conducted between complete and incomplete cases.

We calculated the sample size power of 643 subjects to estimate the prevalence of peripheral arterial disease (PAD) for each criterion using the power command in STATA® V18. We employed the formula for estimating a proportion. We considered expected PAD prevalences based on Schröder's study of 3.7%, 5.4%, and 14.6% for the highest, average, and lowest SAP criteria, respectively. [29] For comparison, the corresponding prevalences observed in our study were 7.8%, 15.4%, and 28.2%. By configuring a two-tailed analysis and a 95% confidence level, he statistical power achieved was 99%, 100%, and 100%, for the highest, average, and lowest SAP criterion respectively.

## Ethical Aspects

This was a retrospective secondary analysis of anonymized medical records. Approval was obtained from the Institutional Ethics and Research Committee of Hospital María Auxiliadora (code HMA/CIEI/015/2024). Informed consent was waived by the committee, as the data were fully anonymized and no identifiable patient information was used or accessed at any point in the study.

## Results

From January 2015 to February 2020, a total of 1,019 patients were enrolled in the at-risk foot program database of María Auxiliadora Hospital. Of the 1,019 patients initially assessed, 376 were excluded: 5 with a history of major amputation, 2 with a history of stroke, and 369 with incomplete ABI data. The final sample comprised 643 participants, evaluated using three ABI calculation methods: highest systolic pressure, average systolic pressure, and lowest systolic pressure. (Fig 2).

### General characteristics

The sample presented a mean age of 61.4 years, of which 69.8% were women. Forty percent of the patients had more than 10 years since their diabetes diagnosis, and 31.9% used insulin alone or in combination with oral antidiabetic drugs as treatment. Among the most important comorbidities, 33.3% had hypertension, 8.6% had a history of ulcer, and 1.4% had a previous coronary artery disease.

Additionally, 28.9% presented peripheral neuropathy, while 61.4% experienced neuropathic pain. Regarding metabolic control targets, only 22.4% had a glycated hemoglobin level below 7%. When evaluating the integrated target of glycated hemoglobin lower than 7%, LDL below 100 mg/dl, and systolic blood pressure below 140 mmHg, only 4.4% met these criteria (Table 1).

### Prevalence of PAD and AC

PAD was present according to the criteria of highest, lowest, and average SAP in 7.8%, 28.2%, and 15.4%, respectively. AC was present in 18.2%, 16.2%, and 11% for the same criterion. AC was assessed using a hierarchical approach: PAD

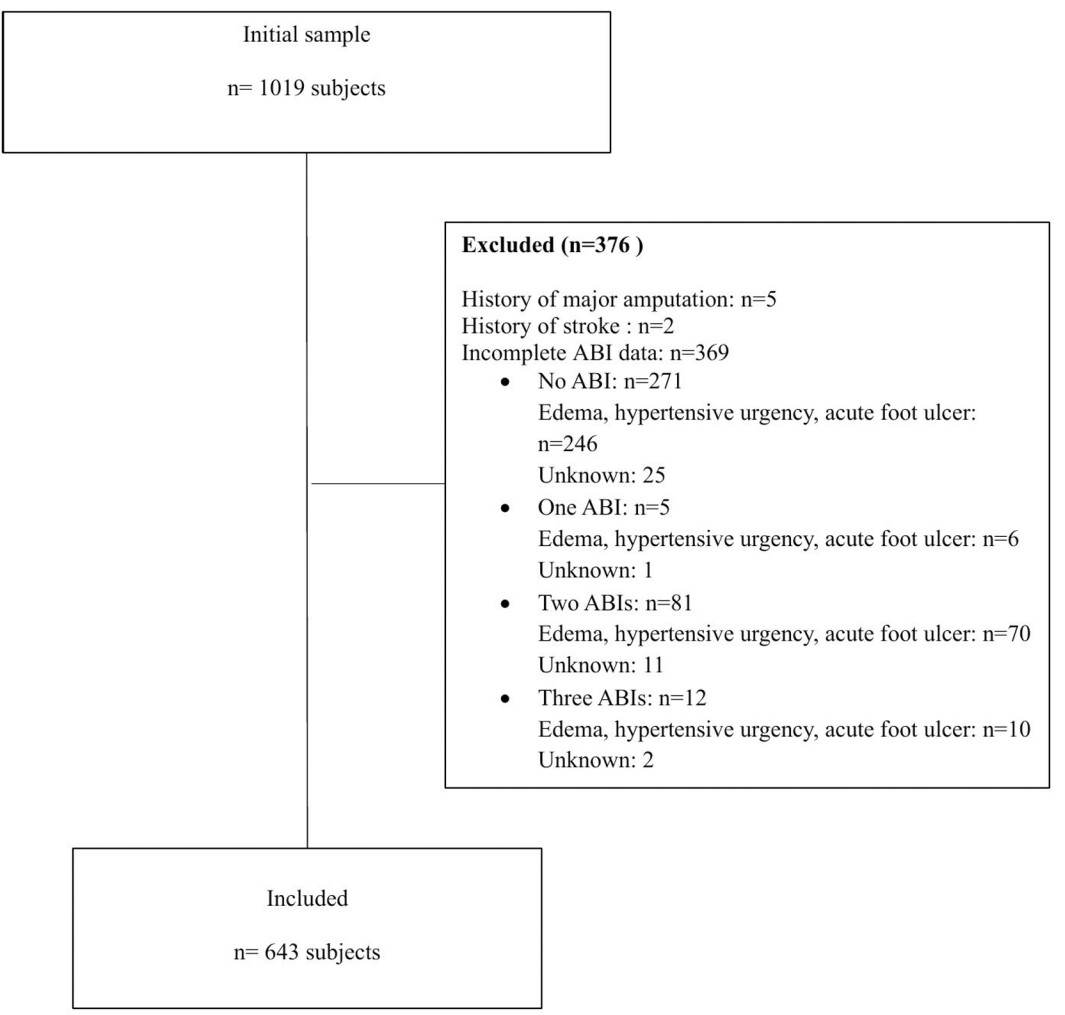

**Fig 2. Flowchart of patients included in the study.**

was identified first, and AC (ABI > 1.3) was diagnosed only in participants without PAD. A small overlap was observed, with 2.0% to 7.7% of PAD cases also meeting AC criteria, depending on the ABI calculation method, indicating that most AC diagnoses were independent of PAD. (Table 2).

To compare PAD and AC classification across ABI calculation methods (highest, lowest, and average SAP), we used symmetry tests for paired categorical data. All pairwise comparisons showed strong evidence of asymmetry (p < 0.001).

S2, S3, and S4 Tables detail the absolute and relative frequencies of each clinical-demographic variable according to the ABI results for each of the three methods. S5 Table describes the means of each quantitative variable according to the ABI results.

## Characteristics associated with PAD

For PAD, age was the only factor associated across all three methods. For the highest SAP method, in subjects over 75 years old, PAD increased by 7.1 times compared to those under 60 years old (aPR 8.2; 95% CI 1.11–60.1). Patients with a history of ulcers showed nearly three times higher prevalence of PAD (aPR 2.88; 95% CI 1.11–7.49; P < 0.05).

**Table 1. Demographic, Clinical, and Laboratory Characteristics of Patients with Diabetes Mellitus Treated in a Peruvian Public Hospital.**

| | Available data N (%) | Value |
|---|---|---|
| **Demographics** | | |
| Male, n(%) | 643 (100) | 194 (30.2) |
| Age (years), Media (SD) | 643 (100) | 61.4 (11.0) |
| Age group, n(%) | 643 (100) | |
|   < 60 | | 291 (43.7) |
|   60 to 74.9 | | 276 (42.9) |
|   ≥ 75 | | 76 (11.8) |
| Education level, n(%) | 643 (100) | |
|   None | | 39 (6.1) |
|   Elementary | | 219 (34.1) |
|   High-School | | 311 (48.4) |
|   College | | 74 (11.5) |
| **Past medical history** | | |
| Diabetes diagnosis time (years), Median [IQR] | 643 (100) | 7 [3–14] |
|   < 10 | | 382 (59.4) |
|   10 to 19.9 | | 180 (28.0) |
|   ≥ 20 | | 81 (12.6) |
| Diabetes treatment, n(%) | 643 (100) | |
|   Not medication | | 34 (5.3) |
|   Oral agents[a] | | 367 (57.1) |
|   Insulin & Oral agents[a] | | 37 (5.8) |
|   Only insulin | | 205 (31.9) |
| Previous diabetic foot ulcer | 643 (100) | 55 (8.6) |
| Diabetic retinopathy | 346 (53.8) | 32 (9.3) |
| Coronary artery disease | 500 (77.8) | 7 (1.4) |
| Hypertension | 408 (63.5) | 136 (33.3) |
| **Clinical evaluation** | | |
| Peripheral neuropathy, n(%)[b] | 643 (100%) | 186 (28.9) |
| Painful neuropathy, n(%) | 643 (100%) | 395 (61.4) |
| Altered foot pulses[b] | 643 (100) | 155 (24.1) |
| BMI (kg/m$^2$), median [IQR] | 425 (66.1) | 27.3 [24.2 to 30.5] |
| BMI category, n(%) | 425 (66.1) | |
|   < 25.0 kg/m$^2$ | | 126 (29.7) |
|   25.0 - 29.9 kg/m$^2$ | | 178 (41.9) |
|   ≥ 30.0 kg/m$^2$ | | 121 (28.4) |
| Systolic brachial pressure (mm Hg), median [IQR] | 643 (100) | 137 [123 a 155] |
| Systolic brachial pressure < 140 mmHg | 643 (100) | 340 (52.9) |
| **Laboratory findings** | | |
| Glucose (mg/dl), median [IQR] | 400 (62.2) | 150 [117 to 196,5] |
| A1c (%), median [QIR] | 325 (50.5) | 9.2 [7.1 a 11.7] |
| A1c < 7%, n(%) | 325 (50.5) | 73 (22.4) |
| Creatinine (mg/dl), median [IQR] | 378 (58.8) | 0.77 [0.63 to 0.98] |
| Estimated glomerular flitration rate < 60 ml/min, n(%)[d] | 379 (58.9) | 85.2 (27.4) |
| Hemoglobin (g/dl), median [IQR] | 338 (52.6) | 13.0 [11.2 a 14.6} |
| HDL-c (mg/dl), median [IQR] | 374 (58.2) | 51 [40 a 67] |

*(Continued)*

**Table 1.** (Continued)

| | Available data N (%) | Value |
|---|---|---|
| LDL-c (mg/dl), median [IQR] | 373 (58.0) | 121 [89 a 151] |
| LDL-c < 100 mg/dl, n(%) | 373 (58.0) | 140 (37.5) |
| Triglycerides, median [IQR] | 357 (55.5) | 151 [114 a 192] |
| Triglycerides < 150 mg/dl | 357 (55.5) | 174 (48.7) |
| Albuminuria (mg/24h), median [IQR] | 276 (42.9) | 10.3 [5.9 a 15.6} |
| Albuminuria 30 mg/24 h, n(%) | | 33 (8.3) |
| A1c < 7%, LDL-c < 100 mg/dl y SBP < 140 mmHg | 315 (49.0) | 14 (4.4) |

IQR:: Interquartile range [25th percentile to 75th percentile]. ABI: Ankle-brachial index. A1c: Glycated hemoglobin. SBP: Systolic brachial pressure. BMI: Body mass index.

[a]Oral agents; (Metformin or glyburide). [b]Michigan Neuropathy Screening Instrument score >2. [c]Positive if there is an absence of pulse in any of the arteries: right or left pedal, right or left posterior tibial. [d]Estimated glomerular filtration rate calculated by the CKD-EPI.

Given the high exclusion rate, we compared included and excluded patients Included patients were older (mean age 61.4 ± 11.0 vs. 59.3 ± 11.0 years; $p = 0.002$), had a higher prevalence of hypertension (33.2% vs. 19.0%; $p < 0.001$) and neuropathy (81.3% vs. 71.8%; $p = 0.001$), and lower mean glucose levels (172.9 ± 80.1 vs. 189.0 ± 87.3 mg/dL; $p = 0.024$) compared with excluded patients. Other demographic, clinical, and laboratory characteristics were similar between groups. (S1 Table).

**Table 2. Prevalence of Peripheral Arterial Disease and Arterial Calcification by ABI Calculation Criteria in Patients with Diabetes Mellitus at a Peruvian Public Hospital.**

| | Highest SAP criterion | | Average SAP criterion | | Lowest SAP criterion | |
|---|---|---|---|---|---|---|
| | N | Prevalence (95% CI) | N | Prevalence (95% CI) | N | Prevalence (95% CI) |
| **All** | | | | | | |
| PAD | 50 | 7.8 (5.8-10.1) | 99 | 15.4 (12.7-18.4) | 181 | 28.2 (24.7-31.7) |
| Only PAD* | 49 | 98.0 | 93 | 93.9 | 167 | 92.3 |
| PAD and AC* | 1 | 2.0 | 6 | 6.1 | 14 | 7.7 |
| Normal | 476 | 74.0 (70.5-77.3) | 473 | 73.6 (69.9-76.9) | 358 | 55.7 (51.7−59−.5) |
| Arterial calcification | 117 | 18.2 (15.3-21.4) | 71 | 11.0 (8.7-13.7) | 104 | 16.2 (13.4-19.3) |
| **Male** | | | | | | |
| PAD | 16 | 8.3 (4.8-13.0) | 28 | 14.4 (9.8-20.2) | 56 | 28.9 (22.6-35.7) |
| Only PAD* | 15 | 93.5 | 25 | 89.3 | 51 | 91.1 |
| PAD and AC* | 1 | 6.3 | 3 | 10.7 | 5 | 8.9 |
| Normal | 141 | 72.7 (65.8-78.8) | 143 | 73.7 (66.9-79.8) | 105 | 54.1 (46.8-61.3) |
| Arterial calcification | 37 | 19.1 (13.8-25.3) | 23 | 11.9 (7.7-17.2) | 34 | 17.0 (12.5-23.6) |
| **Female** | | | | | | |
| PAD | 34 | 7.6 (5.3-10.4) | 71 | 15.8 (12.5-19.6) | 125 | 27.8 (23.7-32.3) |
| Only PAD* | 34 | 100 | 68 | 95.8 | 116 | 92.8 |
| PAD and AC* | 0 | 0 | 3 | 4.2 | 9 | 7.2 |
| Normal | 335 | 74.6 (70.3-78.6) | 330 | 73.5 (69.1-77.5) | 253 | 56.4 (51.6-60.9) |
| Arterial calcification | 80 | 17.8 (14.3-21.7) | 48 | 10.7 (7.9-13.9) | 71 | 15.8 (12.6-19.5) |

ABI: Arm Brachial Index. SAP: Systolic ankle pressure. PAD: Peripheral arterial disease AC: Arterial calcification

*Percentages for "only PAD" and "PAD and AC" refer to the proportion relative to the total number of PAD cases. PAD and AC categories are not mutually exclusive due to the hierarchical classification model.

Conversely, the prevalence of PAD decreased by 88% in obese patients compared to those of normal weight (aPR 0.11; 95% CI 0.03–0.49; p < 0.05). For the lowest SAP method, a significant association with older age was also observed; in those over 75 years, PAD increased by 81% compared to those under 60 years (RP 1.81; 95% CI 1.11–2.95), in conjunction with patients with chronic kidney disease where the prevalence of PAD was 55% (aPR 1.55; 95% CI 1.08–2.23; p < 0.05). For the average SAP criterion, the prevalence increased by 1.87 times (aPR 2.87; 95% CI 1.28–6.43), and the presence of hypertension increased the prevalence of PAD by 87% compared to those without this history (aPR 1.87; 95% CI 1.12–3.13; p < 0.05) (Table 3).

### Characteristics associated with AC

In AC, diabetes duration was the only factor that was associated across all three methods of analysis. In the highest SAP method, individuals with more than 20 years of diabetes had an AC prevalence 2.79 times higher compared to those with less than 10 years (aPR 3.79; 95% CI 2.29–6.25); in the average SAP criterion, the prevalence increased by 2.12 times (aPR 3.12; 95% CI 1.54–6.30; p < 0.001), and for the low ABI, it increased by 2.01 times (aPR 3.01; 95% CI 1.88–4.82; p < 0.001).

Regarding elevated BMI (obesity), in the average SAP method, the prevalence of AC more than doubled compared to subjects with normal weight (aPR 2.26; 95% CI 1.01–5.04; p < 0.05). Similarly, it increased in the lowest SAP criterion (aPR 2.05; 95% CI 1.21–3.45; p < 0.05). For the high ABI, there was a 38% increase, although it was not significant (aPR 1.38; 95% CI 0.76–2.50; p = 0.287).

In patients with a history of ulcers, only in the lowest SAP method, there was double the prevalence of AC compared to those without this history (aPR 2.01, 95% CI 1.24–3.25, p < 0.01). In the other two methods, there was only a 40% increase, which was not significant (Table 4).

## Discussion

Our study found a wide range in the frequency of PAD and a narrow interval of arterial calcification across the three methods used to determine the ABI. Additionally, the highest prevalence of PAD was obtained using the lowest ankle pressure criterion. Conversely, arterial calcification was more frequent when the highest SAP criterion was applied.

### Peripheral arterial disease

There is strong evidence that an ABI less than 0.9 is associated with up to twice the risk of cardiovascular or all-cause mortality in subjects with diabetes. Furthermore, the classic highest SAP criterion is recognized for its better specificity than sensitivity. However, the prevalence of PAD may vary depending on the ABI calculation method applied. Our study's findings on PAD prevalence using various methods are broadly consistent with previous reports. For example, a U.S. study conducted in a multiethnic general population found a prevalence of 3.7%, 5.4% and 14.6% using the highest, average, and lowest SAP criteria, respectively [29]. In contrast, a Frenche primary care cohort reported higher prevalences of 22%, 23%, and 29% for the same criteria [30]. These differences may partly reflect the broader population base and community-level recruitment in those studies, as opposed to our tertiary hospital–based sample of Peruvian adults with type 2 diabetes attending a specialized foot care program, who may present with greater vascular risk. Additionally, methodological variations—such as differences in ABI measurement protocols, cut-off thresholds, and definitions of PAD—may contribute to the higher prevalence observed in our setting. The same US study showed that using different calculation methods can slightly improve prognostic accuracy [29], although such predictive performance was not assessed in our cross-sectional design.

On the other hand, according to the ESVS 2024 guidelines, the lowest SAP criterion is recommended for risk stratification and the highest SAP method is preferred for severity assessment and post-revascularization follow-up [15]. The

**Table 3. PAD Prevalence Ratios by Clinical-Demographic Characteristics Using Three ABI Calculation Criteria.**

| | Highest SAP criterion | | | | Lowest SAP criterion | | | | Average SAP criterion | | | |
|---|---|---|---|---|---|---|---|---|---|---|---|---|
| | Crude PR (95% CI) | p-value | Adjusted PR (95% CI) | p-value | Crude PR (95% CI) | p-value | Adjusted PR (95% CI) | p-value | Crude PR (95% CI) | p-value | Adjusted PR (95% CI) | p-value |
| **Demographics** | | | | | | | | | | | | |
| **Sex** | | | | | | | | | | | | |
| Male | 1.00 | | | | 1.00 | | | | 1.00 | | | |
| Female | 0.90 (0,51−1,59) | 0.726 | | | 0.95 (0,73−1.22) | 0.699 | | | 1.08 (0.72-1.61) | 0.701 | | |
| **Age (years),** | | | | | | | | | | | | |
| < 60.0 | 1.00 | | 1.00 | | 1.00 | | 1.00 | | 1.00 | | 1.00 | |
| 60.0 to 74.9 | 5.91 (2.35-14.8) | **<0.001** | 10.4 (1.59-68.3) | **0.014** | 1.31 (0.99−1.71) | 0.051 | 1.26 (0.81-1.97) | 0.295 | 2.47 (1.57-3.91) | **0.000** | 2.35 (1.71-4.73) | **0.016** |
| ≥ 75.0 | 7.39 (2.66−20.5) | **<0.001** | 8.20 (1.11-60.1) | **0.038** | 1.64 (1.17-2.32) | **0.004** | 1.81 (1.11-2.95) | **0.017** | 2.72 (1.53-4.83) | **0.001** | 2.87 (1.28-6.43) | **0.010** |
| **Education level** | | | | | | | | | | | | |
| None | 1.00 | | | | 1.00 | | | | 1.00 | | | |
| Elementary | 0.83 (0.34- 2.06) | 0.699 | | | 0.85 (0.56-1.31) | 0.476 | | | 0.82 (0.50-1.21) | 0.232 | | |
| High-School | 0.59 (0.23-1.46) | 0.253 | | | 0.76 (0.49−1.15) | 0.195 | | | 0.73 (0.51-1.20) | 0.201 | | |
| College | 0.24 (0.5-1.20) | 0.083 | | | 0.53 (0.29−0.98) | 0.044 | | | 0.59 (0.32-0.92) | 0.048 | | |
| **Past medical history** | | | | | | | | | | | | |
| **Duration of diabetes (years)** | | | | | | | | | | | | |
| < 10.0 | 1.00 | | 1.00 | | 1.00 | | 1.00 | | 1.00 | | 1.00 | |
| 10 to 19.9 | 1.62 (0.87-3.01) | 0.128 | 0.82 (0.35-1.92) | 0.653 | 1.33 (1.02-1.73) | **0.035** | 1.24 (0.83-1.86) | 0.287 | 1.42 (0.94-2.15) | 0.092 | 1.12 (0.61-2.07) | 0.702 |
| ≥ 20.0 | 3.67 (1.96-6.90) | **<0.001** | 1.57 (0.62-3.94) | 0.337 | 1.89 (1.40−2.56) | **<0.001** | 1.39 (0.85-2.26) | 0.179 | 2.22 (1.42-3.47) | **<0.001** | 1.46 (0.74-2.85) | 0.270 |
| **Diabetes treatment** | | | | | | | | | | | | |
| None | 1.00 | | | | 1.00 | | | | 1.00 | | | |
| Oral agentes[a] | 0.58 (0.22-1-52) | 0.268 | | | 0.71 (0.42-1.21) | 0.213 | | | 0.66 (0.31-1.40) | 0.284 | | |
| Insulin & oral agents | 0.48 (0.09-2.41) | 0.374 | | | 0.62 (0.28−1.41) | 0.259 | | | 0.81 (0.29-2.25) | 0.697 | | |
| Insulin only | 0.65 (0.23-1.79) | 0.409 | | | 1.17 (0.70-1.95) | 0.544 | | | 0.94 (0.44-2.01) | 0.881 | | |
| **Previous diabetic foot ulcer** | | | | | | | | | | | | |
| No | 1.00 | | 1.00 | | 1.00 | | 1.00 | | 1.00 | | 1.00 | |
| Yes | 3.30 (1.79−6.06) | **<0.001** | 2.88 (1.11-7.49) | **0.030** | 1.63 (1.18−2.25) | **0.003** | 1.51 (0.89-2.55) | 0.119 | 1.57 (0.93-2.66) | 0.088 | 1.71 (0.82-3.56) | 0.149 |
| **Hypertension** | | | | | | | | | | | | |
| No | 1.00 | | 1.00 | | 1.00 | | 1.00 | | 1.00 | | 1.00 | |

*(Continued)*

Table 3. (Continued)

| | Highest SAP criterion | | | | Lowest SAP criterion | | | | Average SAP criterion | | | |
|---|---|---|---|---|---|---|---|---|---|---|---|---|
| | Crude PR (95% CI) | p-value | Adjusted PR (95% CI) | p-value | Crude PR (95% CI) | p-value | Adjusted PR (95% CI) | p-value | Crude PR (95% CI) | p-value | Adjusted PR (95% CI) | p-value |
| Yes | 1.89 (1.01–3.54) | **0.044** | 1.62 (0.81-3.23) | 0.168 | 1.29 (0.96–1.75) | 0.090 | 1.36 (0.96-1.93) | 0.088 | 1.96 (1.26-3.05) | **0.003** | 1.87 (1.12-3.13) | **0.016** |
| **Clinical evaluation** | | | | | | | | | | | | |
| **Peripheral neuropathy**[b] | | | | | | | | | | | | |
| No | 1.00 | | 1.00 | | 1.00 | | 1.00 | | 1.00 | | 1.00 | |
| Yes | 1.74 (1.02-2.97) | **0.039** | 1.31 (0.56-3.06) | 0.531 | 1.29 (1.02–1.66) | **0.035** | 1.10 (0.74-1.64) | 0.613 | 1.41 (0.97-2.03) | 0.067 | 1.06 (0.61-1.86) | 0.827 |
| **Altered foot pulses**[c] | | | | | | | | | | | | |
| No | 1.00 | | | | 1.00 | | | | 1.00 | | | |
| Yes | 3.16 (1.88-5.32) | **<0.001** | | | 3.00 (2.41- 3.73) | **<0.001** | | | 3.63 (2.56-5.14) | **<0.001** | | |
| **BMI (kg/m²)** | | | | | | | | | | | | |
| < 25.0 | 1.00 | | 1.00 | | 1.00 | | 1.00 | | 1.00 | | 1.00 | |
| 25.0 to 29.9 | 0.51 (0.26–0.97) | **0.041** | 0.36 (0.17-0.76) | **0.008** | 0.94 (0.68–1.28) | 0.706 | 0.81 (0.56-1.18) | 0.282 | 0.66 (0.41-1.05) | 0.082 | 0.57 (0.33-0.97) | **0.041** |
| ≥ 30.0 | 0.11 (0.03–0.49) | **<0.001** | 0.11 (0.01–0.87) | **0.037** | 0.63 (0.41–0.97) | 0.036 | 0.81 (0.51-1.30) | 0.509 | 0.45 (0.23-0.85) | **0.014** | 0.49 (0.23-1.04) | 0.065 |
| **Laboratory findings** | | | | | | | | | | | | |
| **eGFR < 60 mL/min/1.73 m²**[d] | | | | | | | | | | | | |
| No | 1.00 | | 1.00 | | 1.00 | | 1.00 | | 1.00 | | 1.00 | |
| Yes | 2.27 (1.13-4.54) | **0.020** | 1.57 (0.85-2.91) | 0.145 | 1.79 (1.32−2.43) | **<0.001** | 1.55 (1.08-2.23) | **0.015** | 1.91 (1.15-3.17) | **0.012** | 1.43 (0.83-2.49) | 0.194 |
| **A1C (%)** | | | | | | | | | | | | |
| < 7% | 1.00 | | | | 1.00 | | | | 1.00 | | | |
| ≥ 7% | 1.19 (0.45-3.08) | 0.709 | | | 1.15 (0.74–1.79) | 0.516 | | | 2.02 (0.89-4.58) | 0.090 | | |

PAD: Peripheral arterial disease. ABI: Ankle-brachial index. BMI: Body mass. eGFR:Estimated glomerular filtration rate Index PR: Prevalence ratio. 95% CI: 95% confidence interval.

[a]Oral agents; (Metformin or glyburide).[b] Michigan Neuropathy Screening Instrument score >2.[c]. Positive if there is an absence of pulse in any of the arteries: right or left pedal, right or left posterior tibial.[d]. Estimated glomerular filtration rate calculated by the CKD-EPI.

lowest SAP criterion increases prevalence and sensitivity for identifying high-risk patients, though it decreases specificity for diagnosing those with early-stage disease. In practice, diagnosing asymptomatic PAD with an ABI below 0.9 would elevate cardiovascular risk to higher categories. Considering the European cardiovascular risk score for dyslipidemia management (SCORE), with every diabetic patient having at least moderate cardiovascular risk, the presence of asymptomatic PAD would elevate to a high or very high level and intensify risk factor treatment [31].

Regarding factors associated with PAD, our study found a consistent association with older age across all three ABI calculation methods, although the strength of association varied by method. This aligns with international evidence

**Table 4. AC Prevalence Ratios by Clinical-Demographic Characteristics Using Three ABI Calculation Criteria".**

| | Highest SAP criterion | | | | Lowest SAP criterion | | | | Average SAP criterion | | | |
|---|---|---|---|---|---|---|---|---|---|---|---|---|
| | Crude PR (95% CI) | p-value | Adjusted PR (95% CI) | p-value | Crude PR (95% CI) | p-value | Adjusted PR (95% CI) | p-value | Crude PR (95% CI) | p-value | Adjusted PR (95% CI) | p-value |
| **Demographics** | | | | | | | | | | | | |
| **Sex** | | | | | | | | | | | | |
| Male | 1.00 | | | | 1.00 | | | | 1.00 | | | |
| Female | 0.92 (0.65-1.31) | 0.671 | | | 0.91 (0.63-1.31) | 0.636 | | | 0.91 (0.57-1.46) | 0.712 | | |
| **Age (years),** | | | | | | | | | | | | |
| < 60.0 | 1.00 | | 1.00 | | 1.00 | | 1.00 | | 1.00 | | 1.00 | |
| 60.0 to 74.9 | 0.64 (0.45-0.92) | 0.017 | 0.65 (0.41-1.04) | 0.078 | 0.63 (0.43−0.92) | 0.018 | 0.57 (0.37-0.87) | **0.010** | 0.61 (0.38-1.00) | 0.050 | 0.89 (0.51-1.56) | 0.693 |
| ≥ 75.0 | 0.77 (0.44-1.34) | 0.359 | 0.60 (0.29-1.25) | 0.173 | 0.95 (0.55-1.62) | 0.857 | 1.07 (0.59-1.92) | 0.810 | 0.86 (0.42-1.73) | 0.666 | 0.85 (0.32-2.20) | 0.743 |
| **Education level** | | | | | | | | | | | | |
| None | 1.00 | | | | 1.00 | | | | 1.00 | | | |
| Elementary | 1.51 (0.57-3.98) | 0.404 | | | 1.24 (0.48−3.20) | 0.657 | | | 1.38 (0.34-5.62) | 0.648 | | |
| High-School | 1.78 (0.69-4.61) | 0.230 | | | 1.37 (0.54-3.47) | 0.501 | | | 1.94 (0.49-7.62) | 0.341 | | |
| College | 2.00 (0.73-5.51) | 0.177 | | | 1.67 (0.62-4.46) | 0.305 | | | 2.38 (0.56-9.95) | 0.234 | | |
| **Past medical history** | | | | | | | | | | | | |
| **Duration of diabetes (years)** | 1.00 | | 1.00 | | 1.00 | | 1.00 | | 1.00 | | 1.00 | |
| < 10.0 | 1.04 (0.70-1.56) | 0.825 | 1.66 (0.96-2.85) | 0.066 | 1.16 (0.76−1.76) | 0.483 | 1.63 (1.03-2.56) | **0.034** | 1.24 (0.75-2.04) | 0.392 | 2.28 (1.22-4.28) | **0.010** |
| 10.0 to 19.9 | 2.34 (1.61-3.40) | <0.001 | 3.79 (2.29-6.25) | **<0.001** | 2.67 (184−3.87) | <0.001 | 3.01 (1.88-4.82) | **<0.001** | 1.76 (0.98-3.17) | 0.059 | 3.12 (1.54-6.30) | **0.001** |
| ≥ 20.0 | | | | | | | | | | | | |
| **Diabetes treatment** | | | | | | | | | | | | |
| None | 1.00 | | | | 1.00 | | | | 1.00 | | | |
| Oral agents | 0.69 (0.37-1.31) | 0.261 | | | 0.59 (0.32-1.10) | 0.100 | | | 0.51 (0.25-1.32) | 0.364 | | |
| Insulin & oral agents | 1.07 (0.48-2.36) | 0.864 | | | 0.80 (0.35-1.81) | 0.594 | | | 0.51 (0.17.1.57) | 0.246 | | |
| Insulin only | 0.71 (0.37-1.39) | 0.327 | | | 0.76 (0.39-1.43) | 0.392 | | | 0.45 (0.21-1.00) | 0.067 | | |
| **Previous diabetic foot ulcer** | | | | | | | | | | | | |
| No | 1.00 | | 1.00 | | 1.00 | | 1.00 | | 1.00 | | 1.00 | |
| Yes | 2.21 (1.48.3.30) | <0.001 | 1.31 (0.66-2.61) | 0.437 | 2.12 (1.39-3.23) | <0.001 | 2.01 (1.24-3.25) | **0.004** | 1.61 (0.90-3.16) | 0.101 | 1.43 (0.58-3.51) | 0.434 |
| **Hypertension** | | | | | | | | | | | | |
| No | 1.00 | | | | 1.00 | | | | 1.00 | | | |

*(Continued)*

 

Table 4. (Continued)

| | Highest SAP criterion | | | | Lowest SAP criterion | | | | Average SAP criterion | | | |
|---|---|---|---|---|---|---|---|---|---|---|---|---|
| | Crude PR (95% CI) | p-value | Adjusted PR (95% CI) | p-value | Crude PR (95% CI) | p-value | Adjusted PR (95% CI) | p-value | Crude PR (95% CI) | p-value | Adjusted PR (95% CI) | p-value |
| Yes | 1.09 (0.69-1.71) | 0.707 | | | 1.16 (0.73-1.86) | 0.521 | | | 1.63 (0.93-2.87) | 0.085 | | |
| **Clinical evaluation** | | | | | | | | | | | | |
| **Peripheral neuropathy[b]** | | | | | | | | | | | | |
| No | 1.00 | | | | 1.00 | | | | 1.00 | | | |
| Yes | 0.97 (0.68-1.41) | 0.899 | | | 1.06 (0.73−1.55) | 0.740 | | | 1.10 (0.68-1.77) | 0.685 | | |
| **Altered foot pulses[a]** | | | | | | | | | | | | |
| No | 1.00 | | | | 1.00 | | | | 1.00 | | | |
| Yes | 0.78 (0.51-1.21) | 0.275 | | | 1.23 (0.79-1.93) | 0.350 | | | 0.98 (0.56-1.72) | 0.953 | | |
| **BMI (kg/m²)** | | | | | | | | | | | | |
| < 25.0 | 1.00 | | 1.00 | | 1.00 | | 1.00 | | 1.00 | | 1.00 | |
| 25.0 to 29.9 | 1.43 (0.85-2.41) | 0.171 | 1.34 (0.79-2.26) | 0.267 | 1.51 (0.86-2.65) | 0.151 | 1.67 (0.98-2.85) | **0.057** | 1.81 (0.84-3.89) | 0.126 | 1.57 (0.73-3.37) | 0.246 |
| ≥ 30.0 | 1.57 (0.92-2.68) | 0.098 | 1.38 (0.76-2.50) | 0.287 | 1.73 (0.98-3.05) | 0.057 | 2.05 (1.21-3.45) | **0.007** | 2.72 (1.29-5.76) | 0.009 | 2.26 (1.01-5.04) | **0.046** |
| **Laboratory findings** | | | | | | | | | | | | |
| **eGFR<60 mL/min/1.73 m²[d]** | | | | | | | | | | | | |
| No | 1.00 | | | | 1.00 | | | | 1.00 | | | |
| Yes | 1.21 (0.72-2.01) | 0.461 | | | 1.45 (0.86-2.45) | 0.162 | | | 1.36 (0.72-2.56) | 0.329 | | |
| **A1C (%)** | | | | | | | | | | | | |
| < 7.0% | 1.00 | | 1.00 | | 1.00 | | | | 1.00 | | 1.00 | |
| ≥ 7.0% | 1.51 (0.81-2.82) | 0.188 | 1.24 (0.67-2.27) | 0.496 | 1.29 (0.69-2.40) | 0.417 | | | 1.93 (0.86-4.40) | 0.113 | 1.64 (0.73-3.64) | 0.223 |

AC: Arterial calcification. ABI: Ankle-brachial index. BMI: Body mass. eGFR:Estimated glomerular filtration rate Index PR: Prevalence ratio. 95% CI: 95% confidence interval.

[a]Oral agents; (Metformin or glyburide).[b] Michigan Neuropathy Screening Instrument score >2.[c]. Positive if there is an absence of pulse in any of the arteries: right or left pedal, right or left posterior tibial.[d]. Estimated glomerular filtration rate calculated by the CKD-EPI.

indicating that age is one of the strongest independent predictors of PAD in individuals with diabetes, with prevalence increasing steadily in older age groups [32–34]. The increased prevalence observed at advanced ages likely reflects the cumulative burden of atherosclerosis and vascular risk in patients chronically exposed to diabetes [35]. We found no association with gender; recent evidence indicates that women have a similar or even higher prevalence of PAD compared with men, despite historically being considered at lower risk. Differences in presentation, healthcare engagement, and comorbidities may influence detection, underscoring the need for equal screening priority in both sexes [36,37]. Another

risk factor is ethnicity, where Afro-descendants have a higher likelihood. In our study, the original data did not contain this information, but it wouldn't be a limitation given the high degree of mixed ethnicity in our country, unlike other South American countries [38]. An inverse association between obesity and PAD was observed with the highest SAP criterion, consistent with the "obesity paradox" described in previous studies [39]. Potential explanations include confounding by metabolic factors and treatment intensity [40], BMI's limitations as an adiposity measure [41], reverse causality from weight loss due to advanced disease [42], and biological differences in inflammatory or adipose tissue profiles [43]. Given the sole use of BMI and limited subgroup sizes, these results should be interpreted cautiously, and future studies should employ body composition measures and longitudinal designs.

## Arterial calcification

Traditionally, AC was considered a generalized passive process typical of advanced age and metabolic alterations; however, it has recently been discovered as an active and regulated process similar to bone metabolism and thus modifiable. It can occur at the intima or media layer, with both coexisting to different extents [44]. An ABI greater than 1.3 demonstrates the existence of medial or Monckeberg calcification in lower limbs, causing vascular stiffness, possibly with low flow, in an artery that cannot be compressed. It was long considered an artifact, requiring other methods like the toe-brachial index, as finger arteries are less susceptible to AC [45]. However, it has been highlighted as a critical cardiovascular risk marker in the general population [46] and in patients with diabetes [47].

In our study, the prevalence of an ABI > 1.3 in any limb was 18.3%, lower than the 24.4% reported in a Chinese general population study using the average SAP method. However, our hierarchical diagnostic algorithm prioritized PAD over AC, meaning that participants meeting criteria for both conditions ("overlap cases") were classified as PAD. Depending on the ABI calculation method, the proportion of overlap cases ranged from 2.0% (highest ABI) to 7.7% (lowest ABI), which likely led to an underestimation of AC prevalence—particularly in subgroups where both arterial stenosis and calcification are common—and a slight overestimation of PAD prevalence. This reclassification may also have attenuated AC–risk factor associations. Current guidelines recommend cut-off points of either 1.3 [22] or 1.4 [15,23]. for AC diagnosis, with higher thresholds increasing specificity but lowering prevalence. The choice of cut-off should consider epidemiological criteria such as population risk, disease prevalence, and the available resources for confirmatory testing and management. [48]

The most studied factors associated with AC include diabetes mellitus, chronic kidney disease, advanced age, and increased creatinine [49]. In contrast, AC in our study was associated with longer diabetes duration for all calculation methods. This is consistent with findings from the FinnDiane study and other cohorts in Latin America and Asia, which identify disease duration as a key determinant of vascular calcification and PAD [33,50,6] Interestingly, age was not significantly associated with AC in our cohort and even showed a trend toward higher prevalence in younger patients, suggesting that in certain individuals—possibly with specific metabolic or genetic profiles—calcification may develop early in the disease course. In the lowest ABI method, AC was also associated with higher BMI, which may reflect shared pathways between obesity, vascular stiffness, and medial calcification. Chronic kidney disease, although present in approximately one quarter of participants with available data, was not associated with higher AC prevalence, a finding that contrasts with some previous reports and warrants further investigation in larger, longitudinal datasets.

Being a generalized process, arterial calcification (AC) can be assessed using other diagnostic strategies, both invasive (arterial angiography, intravascular ultrasound) and non-invasive (carotid intima-media thickness, brachial artery flow-mediated dilation, or coronary calcification score), which are currently used as risk markers [51].

## Importance of findings

The simultaneous identification of PAD and AC in diabetic patients implies a dual approach to preventing amputations and reducing the risk of cardiovascular events. In our study, the reported prevalence of PAD varied substantially (7.8% to 28.2%) depending on the ABI calculation method, reflecting inherent methodological differences. Using the highest ankle

systolic pressure—as recommended by current guidelines—prioritizes specificity, minimizing false positives by reducing the impact of segmental disease, but may miss cases with isolated arterial lesions, especially in patients with arterial calcification. [23,48] In contrast, the lowest ankle pressure increases sensitivity, detecting more true cases and identifying additional at-risk individuals, but at the expense of more false positives, which could lead to unnecessary further testing, higher healthcare costs, and patient anxiety. [52] The average ankle pressure method produces intermediate prevalence estimates and may be useful in research contexts for prognostic modeling.

From a clinical perspective, the highest ankle pressure approach may be preferable for general population screening, where specificity and comparability with existing literature are critical. Conversely, in high-risk populations such as patients with diabetes, the lowest ankle pressure may be appropriate to avoid missed diagnoses, provided that confirmatory testing is integrated into the diagnostic process. [15]

Furthermore, arterial calcification (AC) should be systematically reported as an additional vascular risk factor during routine screening, rather than being regarded solely as a technical limitation that necessitates further imaging to confirm PAD. Early identification of AC may provide opportunities for targeted preventive interventions, given its potential link to adverse outcomes. In our setting, it remains essential to locally validate the prognostic value of AC—alongside different ABI calculation methods—against hard outcomes such as cardiovascular events, major amputations, and frailty. Such validation would help determine whether incorporating AC detection into routine screening protocols could improve long-term patient management and outcomes, particularly in high-risk diabetic populations. [53]

In contexts such as Peru, where access to specialized vascular assessment is often restricted to urban or referral centers, results may overrepresent patients with more advanced disease or better healthcare access. This underscores the importance of designing population-based studies that capture a broader spectrum of patients and can provide prevalence estimates applicable to the general population. Findings from studies like ours can serve as an important starting point and inform the design, scope, and resource allocation of larger, better-funded epidemiological investigations aimed at strengthening external validity and supporting health policy planning.

## Limitations and strengths of the study

This study has limitations that should be considered when interpreting the findings. First, the hospital-based design, drawn from a tertiary-level Foot at Risk Clinic, may limit generalizability, as the sample may not represent the broader population of patients with type 2 diabetes in other care settings. This could lead to an overestimation of PAD prevalence, as individuals attending specialized tertiary programs may present with more advanced or symptomatic disease compared to community-based or primary care populations. The magnitude of this potential bias could not be quantified due to the lack of comparative data from population-based samples. Attendance at preventive services such as this clinic is often higher among women, which likely explains their predominance in our sample and may partly influence prevalence estimates. Exclusion criteria such as active foot lesions, severe lower-limb pain, poorly controlled blood pressure, or marked edema were applied because these conditions compromise the feasibility and accuracy of ABI measurement in routine clinical practice. However, these exclusions, along with the hierarchical classification algorithm that prioritized PAD over AC, may have introduced selection bias and led to an underestimation of AC prevalence, especially in individuals with asymmetrical disease or concurrent PAD and calcification—potentially attenuating AC–risk factor associations.

Second, a substantial proportion of eligible patients (n = 369, 36.2%) were excluded due to incomplete ABI data, most often related to clinical conditions that impeded measurement. Although the final sample size provided good precision for prevalence estimates, the exclusion of these patients—who may have had more advanced vascular or metabolic disease—could affect the representativeness of the results. We addressed this by comparing included and excluded participants, finding that those excluded were slightly younger, had higher fasting glucose, lower prevalence

of hypertension, and lower prevalence of neuropathy (Supplementary S1 Table), which may have influenced the distribution of outcomes.

Third, only one systolic pressure measurement per artery was obtained, and no formal assessment of intra- or inter-observer variability was performed. While all measurements followed a standardized protocol and were conducted by two endocrinologists with over 10 years of experience in vascular assessment, the absence of repeated measures may introduce measurement bias.

Fourth, some variables had substantial missing data (e.g., HbA1c, chronic kidney disease), and no imputation was performed; complete-case analysis reduced the sample size in multivariable models. Additionally, certain variables (e.g., diabetes duration) were self-reported and may be subject to recall bias, while complications such as diabetic retinopathy may have been underreported due to reliance on self-report rather than active screening.

Finally, some adjusted prevalence ratios—particularly for smaller subgroups—had wide confidence intervals, reflecting statistical imprecision. These estimates should therefore be interpreted cautiously, even though their direction and magnitude are consistent with prior evidence..

A strength of the study is that ABI measurements were performed using a mercury sphygmomanometer and handheld Doppler by trained personnel following a standardized clinical protocol, and peripheral neuropathy was assessed with objective tests, enhancing measurement reliability.

## Conclusions

Using different ABI methods, we observed a prevalence of PAD ranging from 7.8% to 28%, and a prevalence of CA between 11% and 18% among patients with type 2 diabetes mellitus at a public hospital in Peru. The method with the lowest criterion showed the highest prevalence. It is more frequent with increasing age and longer duration of diabetes for any of the three methods. Future research should focus on validating the prognostic utility of these different ABI calculation methods by linking them to hard clinical outcomes such as cardiovascular events, limb ischemia, major amputation, and mortality. In addition, studies are needed to assess the diagnostic accuracy, reproducibility, and cost-effectiveness of each method—particularly in low-resource settings—so as to develop context-specific screening strategies for diabetic populations. Such evidence would help determine whether alternative ABI criteria can improve early detection and targeted prevention of complications in this high-risk group.

## Supporting information

**S1 Table. Comparison of baseline characteristics between included and non-included patients.**
(DOCX)

**S2 Table. Prevalence of PAD and AC According to the Highest SAP Criterion, by Clinical–Demographic Characteristics.**
(DOCX)

**S3 Table. Prevalence of PAD and AC According to the Lowest SAP Criterion, by Clinical–Demographic Characteristics.**
(DOCX)

**S4 Table. Prevalence of PAD and AC According to the Average SAP Criterion, by Clinical–Demographic Characteristics.**
(DOCX)

**S5 Table. Demographic and clinical numerical variables of patients with diabetes mellitus.**
(DOCX)

## Author contributions

**Conceptualization:** Luis Fernando Espinoza-Enciso, Iván Gonzalo Hernández-Gozar, Kevin Clared Zuñiga-Baldarrago, Marlon Yovera-Aldana.

**Data curation:** Luis Fernando Espinoza-Enciso, Kevin Clared Zuñiga-Baldarrago, Marlon Yovera-Aldana.

**Formal analysis:** Iván Gonzalo Hernández-Gozar, Kevin Clared Zuñiga-Baldarrago, Robert Lozano-Purizaca, Manolo Briceño-Alvarado, Marlon Yovera-Aldana.

**Investigation:** Marlon Yovera-Aldana.

**Methodology:** Iván Gonzalo Hernández-Gozar, Kevin Clared Zuñiga-Baldarrago, Manolo Briceño-Alvarado, Marlon Yovera-Aldana.

**Resources:** Luis Fernando Espinoza-Enciso.

**Supervision:** Marlon Yovera-Aldana.

**Validation:** Robert Lozano-Purizaca, Manolo Briceño-Alvarado, Marlon Yovera-Aldana.

**Writing – original draft:** Marlon Yovera-Aldana.

**Writing – review & editing:** Luis Fernando Espinoza-Enciso, Iván Gonzalo Hernández-Gozar, Kevin Clared Zuñiga-Baldarrago, Robert Lozano-Purizaca, Manolo Briceño-Alvarado, Marlon Yovera-Aldana.

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
