## [Decision Letter · Decision Letter 0]

23 Jun 2025

PONE-D-24-58827Different Methods of Ankle-Brachial Index Calculation on the Prevalence of Peripheral Arterial Disease and Arterial Calcification in Subjects with Type 2 Diabetes Mellitus from a Public Hospital in PeruPLOS ONE

Dear Dr. Yovera-Aldana,

Thank you for submitting your manuscript to PLOS ONE. After careful consideration, we feel that it has merit but does not fully meet PLOS ONE’s publication criteria as it currently stands. Therefore, we invite you to submit a revised version of the manuscript that addresses the points raised during the review process.

We look forward to receiving your revised manuscript.

Kind regards,

Yoshihiro Fukumoto

Academic Editor

PLOS ONE

Journal Requirements:

Reviewers' comments:

Reviewer's Responses to Questions

**Comments to the Author**

1. Is the manuscript technically sound, and do the data support the conclusions?

Reviewer #1: Yes

Reviewer #2: Yes

2. Has the statistical analysis been performed appropriately and rigorously? 

Reviewer #1: No

Reviewer #2: Yes

3. Have the authors made all data underlying the findings in their manuscript fully available?

Reviewer #1: Yes

Reviewer #2: Yes

4. Is the manuscript presented in an intelligible fashion and written in standard English?

Reviewer #1: No

Reviewer #2: Yes

5. Review Comments to the Author

Reviewer #1: I am grateful for the opportunity to review the manuscript titled "Different Methods of Ankle-Brachial Index Calculation on the Prevalence of Peripheral Arterial Disease and Arterial Calcification in Subjects with Type 2 Diabetes Mellitus from a Public Hospital in Peru." This cross-sectional study investigates the frequency of peripheral arterial disease (PAD) and arterial calcification (AC) in patients with type 2 diabetes mellitus, utilizing three distinct methods for calculating the ankle-brachial index (ABI) based on data from a hospital-based program in Peru. The objective is to assess how these varying ABI calculation approaches influence the reported prevalence of PAD and AC.

Title:

Page 1, Lines 1-4: The title indicates a cross-sectional study by mentioning "prevalence," but it does not explicitly state the study design (e.g., "A Cross-Sectional Study"). This lack of clarity may obscure the design for readers unfamiliar with prevalence as a hallmark of cross-sectional research.

Page 1, Lines 1-4: The title refers to "Different Methods of Ankle-Brachial Index Calculation" but fails to specify these methods (e.g., highest, lowest, average systolic pressure).

Abstract:

Page 1, Lines 25-48: The abstract omits a background statement, leaving the context and significance of studying PAD and AC in diabetic patients unclear.

Page 1, Lines 29-36: The methods section vaguely describes ABI calculation ("using the lowest, highest, or average systolic pressure") without explaining the numerator and denominator or the measurement process.

Page 1, Lines 37-43: Prevalence rates for PAD and AC are reported, but confidence intervals are not included in the abstract despite being present in the main text (e.g., Page 20, Lines 244-245).

Page 1, Lines 44-48: The conclusions state a prevalence range but misreport "CA" instead of "AC" (likely a typographical error), introducing inconsistency. Additionally, the implications of the findings for clinical practice or research are not articulated beyond a call for further study.

Introduction

Page 10, Lines 52-100: The research question—determining PAD and AC prevalence using different ABI methods—is implied but not explicitly defined.

Page 11, Lines 85-89: The rationale for using three ABI methods is mentioned (e.g., detecting less severe cases or assessing severity), but it lacks justification tied to prior evidence or clinical relevance specific to diabetic patients in Peru.

Page 10, Line 68: The prevalence of PAD in Peruvian hospitals is cited as 18.9% (reference 9), but the cited study focuses on pulsioximetry, not ABI.

Page 12, Lines 101-103: The objectives mention determining PAD and AC frequency using ABI but do not specify the three methods (highest, lowest, average), misaligning with the title and methods section (e.g., Page 13, Lines 121-125).

Methods

Page 13, Lines 105-110: The study is described as cross-sectional, but it does not clarify that it is a secondary analysis of existing data from the Foot at Risk Program.

Page 13, Lines 111-118: Inclusion criteria mention records with complete ABI data, but exclusion criteria (e.g., lesions, severe pain) are listed without explaining their impact on ABI feasibility or prevalence estimates.

Page 17, Lines 208-211: Of 1,019 patients, 369 were excluded for incomplete ABI data, but no reasons (e.g., technical issues, patient factors) are provided.

Page 13, Lines 120-125: ABI methods are defined, but the measurement of systolic pressures (e.g., number of readings, operator consistency) is not detailed.

Page 14, Lines 139-144: Poor metabolic control (HbA1c ≥ 7%) is defined, but its role in the analysis (e.g., as a confounder or outcome modifier) is unclear.

Page 14, Lines 152-155: ABI measurement tools (sphygmomanometer, Doppler) are listed, but the protocol (e.g., cuff size, measurement sequence) and personnel training standardization are not described.

Page 15, Lines 164-165: No imputation for missing data is noted, but the extent and pattern of missingness (e.g., 369 incomplete ABI records) are not analyzed.

Page 13-16: Potential biases (e.g., selection bias from excluding incomplete records, information bias from self-reported variables like diabetes duration) are not identified or addressed.

Page 16, Lines 189-192: Adjustments in Poisson regression are made for variables with p < 0.2, but the rationale for this threshold and the specific confounders adjusted for are not specified.

Page 16, Lines 185-192: Poisson regression with robust variance is used, but the choice over other methods (e.g., logistic regression) and its application (e.g., model assumptions) are not explained.

Page 15, Lines 169-174: The ABI calculation algorithm prioritizes PAD over AC, but this hierarchical approach’s impact on AC prevalence (e.g., underreporting in mixed cases) is not acknowledged.

Results

Page 17, Lines 208-211: The participant flow excludes 369 subjects with incomplete ABI data, but no comparison of excluded vs. included subjects is provided.

Page 18, Lines 219-228: Baseline characteristics (e.g., 69.8% female, mean age 61.4) are reported, but missing data for variables like HbA1c (325/643 records) are not quantified or analyzed for bias.

Page 20, Lines 240-242: Prevalence estimates differ by ABI method (e.g., PAD: 7.8% vs. 28.2%), but the statistical significance of these differences is not tested.

Page 21, Lines 257-269: Adjusted prevalence ratios (aPRs) for PAD (e.g., aPR 8.2 for age > 75) are reported, but confidence intervals are wide (e.g., 1.11-60.1).

Page 25, Lines 280-294: AC associations (e.g., aPR 3.79 for diabetes > 20 years) are presented, but the lack of adjustment for key confounders like age in all models is not justified.

Discussion

Page 29, Lines 304-319: The summary of PAD prevalence aligns with objectives, but the interpretation (e.g., “optimize risk identification”) overstates findings without evidence of diagnostic accuracy.

Page 31, Lines 372-381: The claim that lowest ABI identifies more at-risk individuals assumes a link to mortality outcomes not tested in this study.

Page 32, Lines 389-400: Selection bias from hospital-based sampling and incomplete records is mentioned, but its direction and magnitude (e.g., overestimating PAD prevalence) are not explored.

Page 32, Lines 389-391: Generalizability is questioned due to the hospital setting, but no discussion addresses how the Peruvian context (e.g., healthcare access) affects external validity.

Page 29, Lines 310-319: Comparisons to US and French studies (references 27, 28) are made, but differences in population (e.g., multiethnic vs. Peruvian) and methodology are not critically analyzed.

Page 33, Lines 405-411: Future research is suggested, but the discussion lacks specific gaps (e.g., validation against outcomes) or hypotheses, reducing its utility.

Additional Comments

Throughout Manuscript: Terminology for ABI methods varies (e.g., “highest pressure criterion” on Page 20 vs. “high ABI” on Page 21), risking confusion.

Page 33, Line 407: “CA” is used instead of “AC” (consistent with Abstract, Line 45), indicating a persistent typographical error.

Page 22-28, Tables 3-4: Adjusted models include variables with p < 0.2 from crude analysis, but multicollinearity or model overfitting (e.g., many covariates for 643 subjects) is not addressed.

Reviewer #2: Dear Author

The manuscript presents a cross-sectional study evaluating the prevalence of peripheral arterial disease (PAD) and arterial calcification (AC) in patients with type 2 diabetes mellitus (T2DM) using three different methods for calculating the ankle-brachial index (ABI). The study highlights significant variations in PAD and AC prevalence depending on the ABI calculation method, with important clinical implications. While the study is well-designed and addresses a relevant clinical question, several areas require clarification and improvement.

Major Comments:

Study Design and Methodology:

The manuscript states that the study is observational, cross-sectional, and descriptive. However, it would benefit from a clearer justification for choosing this design over a longitudinal approach, which could provide insights into the progression of PAD and AC over time.

The exclusion criteria (e.g., major amputation, stroke, incomplete ABI data) are appropriate, but the high exclusion rate (369/1019, ~36%) due to incomplete ABIs raises concerns about potential selection bias. Please discuss how this might affect the generalizability of the findings.

ABI Calculation Methods:

The three ABI calculation methods (highest, lowest, and average systolic pressure) are clearly described, but the rationale for selecting these specific methods could be expanded. Are there clinical guidelines or prior studies that support these choices?

The manuscript mentions that the lowest ABI criterion increases sensitivity but decreases specificity. This trade-off should be discussed in greater depth, particularly in terms of clinical decision-making (e.g., how might false positives/negatives impact patient outcomes?).

Prevalence Estimates:

The reported prevalence of PAD varies widely (7.8% to 28.2%) depending on the ABI method. While this variability is a key finding, the discussion should address why such discrepancies exist and which method might be most appropriate for specific clinical scenarios (e.g., screening vs. diagnostic confirmation).

The prevalence of AC (11% to 18.2%) is less variable but still noteworthy. The authors should clarify whether AC was assessed independently of PAD or if overlapping cases were excluded.

Associations with Clinical Variables:

The association of PAD with older age and AC with longer diabetes duration is consistent with prior literature. However, the manuscript could better contextualize these findings by comparing them to similar studies in other populations.

The inverse association between obesity and PAD (using the high ABI method) is intriguing but counterintuitive. The authors should explore potential explanations (e.g., confounding by other metabolic factors) or limitations (e.g., small sample size in subgroups).

Limitations:

The hospital-based design limits generalizability, as acknowledged by the authors. However, the discussion should also address how the study population (e.g., predominantly female, mean age 61.4 years) might influence the results.

The underreporting of AC due to prioritization of PAD in the diagnostic algorithm is a significant limitation. The authors should discuss how this might have affected the observed prevalence and associations.

Minor Comments:

Clarity and Presentation:

The abstract could be more concise. For example, the methodology section repeats details (e.g., ABI calculation methods) that are elaborated in the main text.

Table 1 provides extensive demographic and clinical data, but some variables (e.g., education level, treated hypothyroidism) are not discussed in the results or discussion. Consider streamlining or highlighting the most relevant variables.

Statistical Analysis:

The use of Poisson regression with robust variance is appropriate, but the manuscript should clarify why this method was chosen over logistic regression, especially given the cross-sectional design.

The power calculation is well-described, but it would be helpful to include the actual observed prevalences alongside the expected values for comparison.

Figures and Tables:

Figure 1 (diagnostic flowchart) is clear but could be simplified for readability (e.g., reduce redundant text).

Tables 3 and 4 are dense and could benefit from reorganization (e.g., grouping related variables) to improve clarity.

References:

The references are comprehensive, but some citations (e.g., for ABI cut-off points) could be updated to reflect the most recent guidelines (e.g., 2024 ACC/AHA guidelines cited in the manuscript).

6. PLOS authors have the option to publish the peer review history of their article (what does this mean? ). If published, this will include your full peer review and any attached files.

**Do you want your identity to be public for this peer review?** For information about this choice, including consent withdrawal, please see our Privacy Policy .

Reviewer #1: No

Reviewer #2: No

---

## [Author Response · Author response to Decision Letter 1]

13 Aug 2025

Response to Reviewers – Manuscript PONE-D-24-58827

Manuscript Title: “Prevalence of Peripheral Arterial Disease and Arterial Calcification Based on Three Ankle–Brachial Index Calculation Methods (Highest, Average, and Lowest Systolic Ankle Pressure): A Cross-Sectional Study in Type 2 Diabetes Mellitus Patients in Peru”

Manuscript ID: PONE-D-24-58827

Editorial Requirements

1. Observation: Please ensure that your manuscript meets PLOS ONE's style requirements, including those for file naming.

Response: We have revised and formatted the manuscript and file names to meet PLOS ONE’s editorial requirements, following the provided templates.

2. Observation : Please provide additional details regarding participant consent. In the ethics statement in the Methods and online submission information, please ensure that you have specified (1) whether consent was informed and (2) what type you obtained (for instance, written or verbal, and if verbal, how it was documented and witnessed). If your study included minors, state whether you obtained consent from parents or guardians. If the need for consent was waived by the ethics committee, please include this information.

For additional information about PLOS ONE ethical requirements for human subjects research, please refer to http://journals.plos.org/plosone/s/submission-guidelines#loc-human-subjects-research..

Response: We have updated the Methods section and the submission form with the following statement:

Before: “Approval was obtained from the Institutional Ethics and Research Committee of Hospital María Auxiliadora under code HMA/CIEI/015/2024. The data did not contain any personally identifiable information.”

After: "This was a retrospective secondary analysis of anonymized medical records. Approval was obtained from the Institutional Ethics and Research Committee of Hospital María Auxiliadora (code HMA/CIEI/015/2024). Informed consent was waived by the committee, as the data were fully anonymized and no identifiable patient information was used or accessed at any point in the study."

Reviewer #1 Comments and Responses

3. Observation: Page 1, Lines 1-4: The title indicates a cross-sectional study by mentioning "prevalence," but it does not explicitly state the study design (e.g., "A Cross-Sectional Study"). This lack of clarity may obscure the design for readers unfamiliar with prevalence as a hallmark of cross-sectional research.

Page 1, Lines 1-4: The title refers to "Different Methods of Ankle-Brachial Index Calculation" but fails to specify these methods (e.g., highest, lowest, average systolic pressure).

Response: The title has been revised to include the design and ABI methods calculation:

Before: “Different Methods of Ankle-Brachial Index Calculation on the Prevalence of Peripheral Arterial Disease and Arterial Calcification in Subjects with Type 2 Diabetes Mellitus from a Public Hospital in Peru ”

After: “Prevalence of Peripheral Arterial Disease and Arterial Calcification Based on Three Ankle-Brachial Index Calculation Methods (Highest, Average, and Lowest Systolic Ankle Pressure): A cross-sectional study in Type 2 Diabetes Mellitus Patients in Peru”

4. Observation: Page 1, Lines 25-48: The abstract omits a background statement, leaving the context and significance of studying PAD and AC in diabetic patients unclear.

Page 1, Lines 29-36: The methods section vaguely describes ABI calculation ("using the lowest, highest, or average systolic pressure") without explaining the numerator and denominator or the measurement process.

Page 1, Lines 37-43: Prevalence rates for PAD and AC are reported, but confidence intervals are not included in the abstract despite being present in the main text (e.g., Page 20, Lines 244-245).

Page 1, Lines 44-48: The conclusions state a prevalence range but misreport "CA" instead of "AC" (likely a typographical error), introducing inconsistency. Additionally, the implications of the findings for clinical practice or research are not articulated beyond a call for further study.

Response:

We thank the reviewer for these detailed comments regarding the abstract. In response, we have revised the abstract to include a concise background statement highlighting the clinical relevance of studying peripheral arterial disease (PAD) and arterial calcification (AC) in patients with diabetes, particularly considering the heterogeneity introduced by different ankle-brachial index (ABI) calculation methods.

The methods section of the abstract now explicitly defines how the ABI was calculated, specifying the use of the highest brachial systolic pressure as the denominator, and the highest, lowest, or average systolic ankle pressures (from either dorsalis pedis or posterior tibial arteries) as the numerator. We also clarified that these methods were compared in a population of Peruvian adults with type 2 diabetes.

To improve the precision of reporting, we added 95% confidence intervals for the prevalence estimates of PAD and AC across the three ABI calculation methods. The typographical error "CA" has been corrected to "AC" to maintain consistency.

Finally, the conclusions have been expanded to better reflect the clinical and research implications of the findings, emphasizing the impact of ABI calculation methods on disease classification and the importance of method selection in clinical and epidemiological settings.

5. Page 10, Lines 52-100: The research question—determining PAD and AC prevalence using different ABI methods—is implied but not explicitly defined.

We thank the reviewer for this valuable observation. In response, we have revised the final paragraph of the Introduction to explicitly define the research question. The paragraph now reads: “Therefore, the objective of this study was to determine how the prevalence of peripheral arterial disease (PAD) and arterial calcification (AC) varies according to the method used to calculate the ankle-brachial index (ABI)—specifically, using the highest, average, or lowest ankle systolic pressure—in patients with type 2 diabetes mellitus at Hospital María Auxiliadora, a public hospital in Peru.”

This clarification ensures that the research question is explicitly stated and fully aligned with the study title, methods, and analytical framework.

Therefore, this study aims to determine how the prevalence of peripheral arterial disease and arterial calcification varies according to the method used to calculated the ankle-brachial index (highest, average or lowest ankle systolic pressure) in patients with type 2 diabetes mellitus at a public hospital in Peru..

6. Page 11, Lines 85-89: The rationale for using three ABI methods is mentioned (e.g., detecting less severe cases or assessing severity), but it lacks justification tied to prior evidence or clinical relevance specific to diabetic patients in Peru.

Response: We have incorporated additional references and explanations to support the clinical relevance of using different ABI methods.

“These approaches aim to balance diagnostic sensitivity and specificity while enhancing the prediction of cardiovascular risk and lower-limb function. Although the highest ankle pressure remains the standard for population studies due to its specificity and comparability, the lowest pressure improves sensitivity for detecting milder or asymptomatic cases. The average pressure has shown stronger associations with functional performance, such as walking speed and leg strength. Each method yields different PAD prevalence estimates and identifies distinct risk profiles.”

7. Page 10, Line 68: The prevalence of PAD in Peruvian hospitals is cited as 18.9% (reference 9), but the cited study focuses on pulsioximetry, not ABI..

Response: We have corrected the citation to clarify that the data were based on pulsioximetry and not ABI.

8. Page 12, Lines 101-103: The objectives mention determining PAD and AC frequency using ABI but do not specify the three methods (highest, lowest, average), misaligning with the title and methods section (e.g., Page 13, Lines 121-125).

We thank the reviewer for this valuable observation. In response, we have revised the objectives in both the Abstract and the Introduction to explicitly state that the aim of the study was to determine the frequency of peripheral arterial disease (PAD) and arterial calcification (AC) using three different methods of ABI calculation: the highest, lowest, and average systolic ankle pressures. This modification ensures alignment with the study title, methodology, and analytical approach.

9. Page 13, Lines 105-110: The study is described as cross-sectional, but it does not clarify that it is a secondary analysis of existing data from the Foot at Risk Program.

We thank the reviewer for this helpful observation. We agree with the need to clarify the data source and have updated the Study Design subsection of the Methods section accordingly. The current version now explicitly states that this is a secondary analysis of de-identified clinical data collected prospectively through the Foot at Risk Program at Hospital María Auxiliadora.

10. Page 13, Lines 111-118: Inclusion criteria mention records with complete ABI data, but exclusion criteria (e.g., lesions, severe pain) are listed without explaining their impact on ABI feasibility or prevalence estimates.

We thank the reviewer for this important observation. We have clarified in the revised Methods section that exclusion criteria such as active foot lesions, severe lower-limb pain, poorly controlled blood pressure, or marked edema were applied because these conditions compromise the feasibility and accuracy of ABI measurement in routine clinical practice. However, we also acknowledge that these exclusions may introduce selection bias, as they could systematically exclude patients with more advanced vascular or metabolic disease. This, in turn, may lead to an underestimation of the true prevalence of peripheral arterial disease (PAD) or arterial calcification (AC). We have now explicitly discussed this potential limitation in both the Population and Sample subsection and the Limitations section of the manuscript.

11. Page 17, Lines 208-211: Of 1,019 patients, 369 were excluded for incomplete ABI data, but no reasons (e.g., technical issues, patient factors) are provided.

We thank the reviewer for this comment. We have now detailed the reasons for incomplete ABI data in the revised manuscript. Of the 1,019 patients initially assessed, 376 were excluded: 5 due to a history of major amputation and 2 due to a history of stroke. The remaining 369 were excluded because of incomplete ABI data. Among these, 271 patients had no ABI measurement, mostly due to edema, hypertensive urgency, or acute foot ulcer (n=246), with 25 cases undocumented. Five patients had only one ABI, with reasons documented in 4 cases (clinical conditions as above) and unknown in 1 case. Eighty-one patients had two ABIs, with 70 due to clinical conditions and 11 undocumented. Twelve patients had three ABIs, with 10 due to clinical conditions and 2 undocumented. These details have been added to the Methods section for clarity.

12. Page 13, Lines 120-125: ABI methods are defined, but the measurement of systolic pressures (e.g., number of readings, operator consistency) is not detailed.

We appreciate the reviewer’s insightful comment and have revised the Methods section to provide greater detail. As part of standard clinical practice within the At-Risk Foot Program, only one systolic pressure reading per artery was obtained during each evaluation. Although repeated measurements were not performed and no formal assessment of intra- or inter-observer variability was conducted, all measurements were carried out by two endocrinologists with over 10 years of clinical experience in vascular evaluation, which contributes to consistency and reliability of the measurements. We also acknowledge this limitation and have explicitly added the absence of repeated measurements as a potential source of measurement bias in the study’s Limitations section.

13. Page 14, Lines 139-144: Poor metabolic control (HbA1c ≥ 7%) is defined, but its role in the analysis (e.g., as a confounder or outcome modifier) is unclear.

We thank for this observation. However, given the descriptive and cross-sectional nature of the study, poor metabolic control (defined as HbA1c ≥ 7%) was not modeled as a confounder or effect modifier, since the objective was not to assess causal relationships. Rather, glycemic control was included as a relevant clinical characteristic to describe the population and explore its association with the distribution of ABI categories. We acknowledge that causal inference would require longitudinal data and temporality, which are beyond the scope of this analysis.

14. Page 14, Lines 152-155: ABI measurement tools (sphygmomanometer, Doppler) are listed, but the protocol (e.g., cuff size, measurement sequence) and personnel training standardization are not described.

We thank for this observation. In response, we have expanded the manuscript to include a brief protocol describing how ABI measurements were performed. All assessments followed a standardized clinical procedure: patients rested in the supine position for at least 10 minutes prior to measurement; appropriate cuff sizes were selected based on limb circumference; and systolic pressures were recorded sequentially at both brachial arteries and at the dorsalis pedis and posterior tibial arteries of both ankles. The highest brachial pressure was used as the denominator, and each ankle artery was measured once, in accordance with routine clinical practice. Although no formal intra- or inter-observer agreement assessment was performed, all measurements were conducted by two endocrinologists with over 10 years of experience in vascular and diabetic foot evaluation, which ensures procedural consistency. These additions have been incorporated into the revised Methods section.

15. Page 15, Lines 164-165: No imputation for missing data is noted, but the extent and pattern of missingness (e.g., 369 incomplete ABI records) are not analyzed.

We thank the reviewer for this important observation. As noted, no imputation was performed for missing data, and records with incomplete ABI measurements (n = 369) were excluded from the analysis. While the reasons for missing ABI data were not systematically recorded—likely including technical limitations, patient discomfort, or incomplete documentation—we acknowledge that not characterizing these excluded cases may introduce selection bias. We have now added this point explicitly to the Limitations section, noting that the absence of an analysis of excluded individuals limits our understanding of the extent and potential pattern of missingness and may affect the generalizability of the findings

16. Page 13-16: Potential biases (e.g., selection bias from excluding incomplete records, information bias from self-reported variables like diabetes duration) are not identified or addressed.

We thank the reviewer for this insightful comment. In response, we have revised the Limitations section to explicitly address potential sources of bias. First, we acknowledge the possibility of selection bias due to the exclusion of patients with incomplete ABI data, which may have led to an underrepresentation of individuals with more advanced disease or technical challenges in ABI measurement. Second, we recognize the

---

## [Decision Letter · Decision Letter 1]

31 Aug 2025

Prevalence of Peripheral Arterial Disease and Arterial Calcification Based on Three Ankle-Brachial Index Calculation Methods (Highest, Average, and Lowest Systolic Ankle Pressure): A cross-sectional study in Type 2 Diabetes Mellitus Patients in Peru

PONE-D-24-58827R1

Dear Dr. Yovera-Aldana,

We’re pleased to inform you that your manuscript has been judged scientifically suitable for publication and will be formally accepted for publication once it meets all outstanding technical requirements.

Kind regards,

Yoshihiro Fukumoto

Academic Editor

PLOS ONE

Additional Editor Comments (optional):

Reviewer #2:

Reviewers' comments:

Reviewer's Responses to Questions

**Comments to the Author**

1. If the authors have adequately addressed your comments raised in a previous round of review and you feel that this manuscript is now acceptable for publication, you may indicate that here to bypass the “Comments to the Author” section, enter your conflict of interest statement in the “Confidential to Editor” section, and submit your "Accept" recommendation.

Reviewer #2: All comments have been addressed

2. Is the manuscript technically sound, and do the data support the conclusions?

Reviewer #2: Yes

3. Has the statistical analysis been performed appropriately and rigorously? 

Reviewer #2: Yes

4. Have the authors made all data underlying the findings in their manuscript fully available?

Reviewer #2: Yes

5. Is the manuscript presented in an intelligible fashion and written in standard English?

Reviewer #2: Yes

6. Review Comments to the Author

Reviewer #2: Dear Author

This response document is excellent and demonstrates that the authors have made substantial improvements to the manuscript. The revisions align with the reviewers' comments and enhance the manuscript's scientific quality, clarity, and transparency.

The manuscript is now much stronger and should be considered for acceptance

7. PLOS authors have the option to publish the peer review history of their article (what does this mean? ). If published, this will include your full peer review and any attached files.

**Do you want your identity to be public for this peer review?** For information about this choice, including consent withdrawal, please see our Privacy Policy .

Reviewer #2: No

---

## [Editor Report · Acceptance letter]

PONE-D-24-58827R1

PLOS ONE

Dear Dr. Yovera-Aldana,

I'm pleased to inform you that your manuscript has been deemed suitable for publication in PLOS ONE. Congratulations! Your manuscript is now being handed over to our production team.

Kind regards,

on behalf of

Dr. Yoshihiro Fukumoto

Academic Editor

PLOS ONE